# BDGS-SLAM: A Probabilistic 3D Gaussian Splatting Framework for Robust SLAM in Dynamic Environments

**DOI:** 10.3390/s25216641

**Published:** 2025-10-30

**Authors:** Tianyu Yang, Shuangfeng Wei, Jingxuan Nan, Mingyang Li, Mingrui Li

**Affiliations:** 1School of Geomatics and Urban Spatial Informatics, Beijing University of Civil Engineering and Architecture, Beijing 102616, China; 202306030102@stu.bucea.edu.cn (T.Y.); 202306020130@stu.bucea.edu.cn (J.N.); 2Research Center of Representative Building and Architectural Heritage Database, Ministry of Education, Beijing 102616, China; 3Beijing Key Laboratory for Architectural Heritage Fine Reconstruction & Health Monitoring, Beijing 102616, China; 4School of Computer and Artificial Intelligence, Beijing Technology and Business University, Beijing 102400, China; 2307010207@st.btbu.edu.cn; 5School of Information and Communication Engineering, Dalian University of Technology (DUT), Dalian 116024, China

**Keywords:** 3D Gaussian Splatting, Visual SLAM, Bayesian filtering, dynamic environments, adaptive optimization

## Abstract

Simultaneous Localization and Mapping (SLAM) utilizes sensor data to concurrently construct environmental maps and estimate its own position, finding wide application in scenarios like robotic navigation and augmented reality. SLAM systems based on 3D Gaussian Splatting (3DGS) have garnered significant attention due to their real-time, high-fidelity rendering capabilities. However, in real-world environments containing dynamic objects, existing 3DGS-SLAM methods often suffer from mapping errors and tracking drift due to dynamic interference. To address this challenge, this paper proposes BDGS-SLAM—a Bayesian Dynamic Gaussian Splatting SLAM framework specifically designed for dynamic environments. During the tracking phase, the system integrates semantic detection results from YOLOv5 to build a dynamic prior probability model based on Bayesian filtering, enabling accurate identification of dynamic Gaussians. In the mapping phase, a multi-view probabilistic update mechanism is employed, which aggregates historical observation information from co-visible keyframes. By introducing an exponential decay factor to dynamically adjust weights, this mechanism effectively restores static Gaussians that were mistakenly culled. Furthermore, an adaptive dynamic Gaussian optimization strategy is proposed. This strategy applies penalizing constraints to suppress the negative impact of dynamic Gaussians on rendering while avoiding the erroneous removal of static Gaussians and ensuring the integrity of critical scene information. Experimental results demonstrate that, compared to baseline methods, BDGS-SLAM achieves comparable tracking accuracy while generating fewer artifacts in rendered results and realizing higher-fidelity scene reconstruction.

## 1. Introduction

At present, 3D Gaussian Splatting (3DGS) technology has attracted wide attention for its excellent performance in realistic and high-fidelity scene reconstruction, which has led to the emergence of various 3DGS-based SLAM systems [1,2,3,4,5,6]. For applications such as augmented reality (AR), virtual reality (VR), and robot navigation, SLAM systems with real-time high-fidelity dense mapping capabilities are critical. However, existing representative 3DGS-SLAM schemes, such as Gaussian Splatting SLAM [4] and SplaTAM [6], have significant limitations when dealing with real-world dynamic environments. The ubiquitous presence of moving objects in real-world scenes usually prompts traditional SLAM systems to adopt semantic-based policies for interference rejection [7,8,9,10,11,12,13,14,15]. Unfortunately, such traditional methods based on feature removal or optical flow segmentation are difficult to apply directly to the current 3DGS-SLAM framework. This is because 3DGS maintains a dense, view-dependent Gaussian representation that integrates all observations into the map; dynamic objects introduce inconsistent measurements across views, which manifest as rendering artifacts such as ghosting, blurred textures, or duplicated structures in the reconstructed scene. While feature-based filtering can alleviate local tracking errors, it cannot prevent these global artifacts, since the Gaussian representation continuously aggregates noisy inputs from dynamic regions. As a result, the rendering artifact problem caused by dynamic noise input remains unresolved under traditional strategies.

Recently, SLAM systems based on neural implicit representation (DN-SLAM [16], NID-SLAM [17], DDN-SLAM [18], and RoDyn-SLAM [19]) have made some progress in coping with dynamic interference. Nevertheless, neural implicit methods still face many challenges. For example, DN-SLAM and DDN-SLAM rely mainly on traditional strategies; although DDN-SLAM introduces penalty mechanisms in backend rendering to mitigate artifacts, it fails to completely solve the artifact problem of multi-view synthesis. NID-SLAM and RoDyn-SLAM use optical-flow-based estimation for motion tracking, where RoDyn-SLAM estimates camera pose by eliminating sampled rays with dynamic labels and combining warp loss. However, these methods are still insufficient in terms of their real-time performance and scene geometric accuracy, and their tracking and mapping modules are usually loosely coupled.

To address these challenges, this paper proposes BDGS-SLAM, a 3DGS-SLAM framework optimized for dynamic environments. The framework combines three modules in a unified manner. First, a Bayesian filtering mechanism continuously evaluates the probability of each Gaussian point being static or dynamic, mitigating the influence of segmentation noise and misclassification. Second, a multi-view fusion strategy integrates observations across keyframes with temporal decay, so that the map remains temporally consistent while adapting to scene changes. Third, an adaptive optimization scheme softly penalizes unreliable dynamic points instead of discarding them, thereby preserving valid static structures and avoiding rendering artifacts. Together, these components enable BDGS-SLAM to achieve robust trajectory estimation and faithful scene reconstruction in highly dynamic environments. Experimental validation on several real-world datasets shows that BDGS-SLAM achieves the state-of-the-art level in rendering accuracy and running efficiency.

The main contributions of this paper can be summarized as follows:(1)We propose BDGS-SLAM, a 3DGS-SLAM framework for dynamic scenes that tightly couples tracking and mapping, using Bayesian filtering to refine dynamic/static labels and maintain system-wide consistency.(2)A Bayesian filtering-based dynamic point culling method is introduced to estimate the dynamic probability of Gaussians over time, reducing false removal of static points and preserving map completeness under transient occlusion.(3)A multi-view probability updating mechanism is proposed to refine Gaussian labels using co-visible keyframes with exponential decay weighting, recovering misclassified static points and preventing rendering artifacts.(4)Experimental results on real datasets show that the proposed method achieves optimal current performance in tracking accuracy and rendering quality.

## 2. Related Work

### 2.1. Traditional SLAM

Simultaneous Localization and Mapping (SLAM) has been a foundational problem in robotics and computer vision for over two decades. Traditional SLAM methods assume a static environment and rely on geometric constraints to jointly estimate the camera trajectory and map the scene. Classical systems can be broadly categorized into feature- based and direct methods. Feature-based SLAM, exemplified by ORB-SLAM [20] and ORB-SLAM2 [21], detects and matches sparse visual features across frames, followed by bundle adjustment to optimize poses and landmarks. These systems are robust to viewpoint changes and efficient in well-textured environments.

On the other hand, direct methods such as LSD-SLAM [22] and DVO-SLAM [23] bypass feature extraction and minimize photometric error directly over pixel intensities.This allows them to operate in low-texture regions and under motion blur, making them suitable for semi-dense or dense tracking and mapping. DVO-SLAM further utilizes depth information and dense alignment, improving robustness in indoor environments where depth sensors are available.

Despite their success, most early SLAM systems were designed under the static-world assumption, where observations are treated as if they originate from a rigid, unchanging world. This assumption breaks down in real-world scenarios, where dynamic elements such as humans, vehicles, or animals are ubiquitous. These moving entities introduce inconsistent observations, causing tracking drift, erroneous loop closures, and corrupted map points. Since traditional SLAM lacks mechanisms for motion segmentation or semantic reasoning, it often misinterprets dynamic objects as part of the static scene.

Moreover, these methods rely purely on geometric consistency and ignore high-level semantics or learned priors. As a result, they perform poorly in visually ambiguous environments (e.g., textureless surfaces, repetitive patterns), and their maps are often sparse or semi-dense, limiting their utility in tasks requiring high-fidelity reconstructions or 3D understanding.

Monocular SLAM systems face an additional challenge: scale ambiguity. Without depth input, monocular systems can only recover structure up to an unknown scale. Techniques like scale-aware initialization or loop closure can alleviate this issue to some extent, but scale drift remains a persistent problem, particularly in long trajectories or dynamic scenes.

To support diverse applications, extensions of traditional SLAM have incorporated various sensors and modules. For instance, ORB-SLAM3 [24] unifies monocular, stereo, and RGB-D inputs in a single framework and supports visual–inertial fusion. However, its performance in dynamic environments remains limited due to the absence of motion modeling or dynamic filtering.

Traditional SLAM pipelines such as ORB-SLAM and LSD-SLAM were originally designed under the static scene assumption, where all observations are treated as originating from a rigid, unchanging world. This assumption simplifies optimization but limits robustness in highly dynamic environments. Nevertheless, it is important to note that several extensions of traditional SLAM have been proposed to explicitly handle dynamic objects. For example, DynaSLAM [7] integrates semantic segmentation and motion consistency checks to filter out dynamic regions. Therefore, rather than a strict opposition between “traditional” and “dynamic” SLAM, we adopt the terminology here to emphasize whether a method *explicitly models dynamic objects* or relies on the static-world assumption.

### 2.2. Dynamic SLAM

Dynamic SLAM aims to improve robustness in real-world environments by addressing the challenges posed by moving objects and non-rigid deformations. Traditional SLAM methods often fail in such scenarios due to the static-world assumption. To mitigate this issue, dynamic SLAM systems introduce strategies for detecting, filtering, or modeling dynamic regions. Early approaches primarily rely on semantic segmentation to identify and exclude potentially moving entities from the SLAM pipeline.

DynaSLAM is one of the earliest and most influential works in this direction, integrating Mask R-CNN into the ORB-SLAM2 framework to mask out dynamic objects such as humans and vehicles. By combining semantic segmentation with multi-view geometry, DynaSLAM enables the identification of both a priori dynamic objects and those detected through scene inconsistency. Detect-SLAM [25] follows a similar idea but adopts lightweight object detectors for real-time performance, offering a better trade-off between speed and accuracy. These systems significantly reduce the negative impact of moving objects on pose estimation and map construction by filtering out segmented regions. However, their effectiveness highly depends on the accuracy of the semantic segmentation network and predefined dynamic classes. If a moving object is not part of the segmentation class or is incorrectly segmented, it may introduce severe tracking errors or corrupt the generated map.

Beyond semantic filtering, geometric-based dynamic SLAM systems utilize motion cues derived from scene flow, frame-to-frame residuals, or optical flow fields. DS-SLAM [9] integrates optical flow and feature tracking to separate dynamic and static components in the input frames. By computing the inconsistency between tracked features and optical flow predictions, it effectively filters out dynamic keypoints and maintains reliable SLAM tracking. Co-Fusion [26] employs real-time RGB-D segmentation to update the global map with only static elements while retaining dynamic segments in a separate object-level representation. These methods demonstrate improved adaptability in dynamic environments, particularly when semantic priors are unavailable or unreliable. However, a common limitation is their tendency to discard large amounts of image data, including occluded static backgrounds, resulting in sparse reconstructions and reduced map completeness.

Recently, several neural implicit SLAM approaches have emerged that directly incorporate dynamic modeling into end-to-end optimization frameworks. Methods such as DN-SLAM [16], NID-SLAM [17], and DDN-SLAM [18] use neural signed distance functions (SDFs) or neural volumetric fields to jointly optimize scene structure and camera poses via differentiable rendering. RoDyn-SLAM [19] introduces ray-based residual modeling and jointly estimates dynamic masks and depth using motion segmentation and photometric error feedback. These methods exhibit strong reconstruction potential in dynamic scenes and can adaptively refine the scene geometry. However, the reliance on ray marching and MLP inference incurs significant computational overhead. Moreover, many of them retain a loosely coupled architecture—separating tracking from mapping updates—which limits the system’s ability to respond promptly to dynamic changes.

DG-SLAM [27] represents a more recent attempt to integrate 3D Gaussian Splatting into real-time SLAM for dynamic scenes. DG-SLAM introduces Gaussian-level dynamic labeling using temporal consistency and classifies each Gaussian as static or dynamic. However, its binary labeling mechanism is deterministic and lacks mechanisms for label refinement over time.

To overcome these limitations, several recent studies have explored integrating probabilistic inference and temporal fusion into dynamic SLAM. For example, approaches inspired by Bayesian filtering attempt to maintain belief distributions over the dynamic state of map elements. Rather than hard assigning labels to features, these methods accumulate multi-view evidence across time, offering more robust decisions about whether a point belongs to a dynamic object. Such strategies reduce the risk of misclassification from transient occlusions or partial observations. However, incorporating probabilistic models into high-speed SLAM pipelines introduces additional computational challenges, especially in terms of maintaining temporal coherence and integrating evidence in a scalable manner.

Overall, current dynamic SLAM methods can be broadly divided into two categories: segmentation-based filtering methods and reconstruction-centric implicit methods. The former ensures tracking stability by eliminating dynamic content but at the cost of map completeness. The latter provides high-fidelity map reconstructions but suffers from slow convergence, high memory usage, and limited real-time applicability. Thus, designing an efficient and robust dynamic SLAM framework remains an open and critical challenge, particularly for applications in autonomous driving, augmented reality, and human–robot interaction in unconstrained environments.

### 2.3. Three-Dimensional Gaussian Splatting and NeRF-Based SLAM

Recent advances in neural scene representation have sparked significant interest in integrating implicit and explicit neural rendering models into SLAM pipelines. Neural Radiance Field (NeRF) [28] opened up new possibilities for photorealistic view synthesis, which inspired many dense SLAM frameworks such as ESLAM [29], Co-SLAM [30], iMAP [31], NICE-SLAM [32], Vox-Fusion [33], and Point-SLAM [34]. These methods jointly optimize camera poses and scene geometry using ray-based volumetric rendering. However, due to the high computational cost of ray sampling and MLP inference, they often require depth priors or render sparse pixels for tractability, limiting their generalization and real-time performance.

Beyond single-agent settings, several works extend neural SLAM toward collaborative scenarios. MCN-SLAM [35] enables distributed agents to share scene reconstructions with low communication overhead, while MNE-SLAM [36] achieves real-time multi-agent mapping through cross-agent feature alignment. PLG-SLAM [37] improves global consistency via progressive bundle adjustment, and NeSLAM [38] enhances stability through depth completion and denoising. Although designed for multi-robot collaboration rather than dynamic scenes, these studies underline that scalability and robustness in SLAM benefit from probabilistic and modular designs, an idea also reflected in our BDGS-SLAM framework. Building on these insights, we next introduce BDGS-SLAM, a framework designed specifically for dynamic environments.

In contrast to NeRF-based pipelines, 3D Gaussian Splatting (3DGS) [39] provides an efficient and differentiable rendering mechanism using explicit anisotropic Gaussians. GS-SLAM [1] first introduced 3DGS into SLAM, enabling RGB-D mapping with adaptive Gaussian expansion. Photo-SLAM [2] extended the framework to various camera modalities and improved map fidelity with geometry-based densification and pyramid learning. MonoGS [4] explored monocular SLAM with 3DGS, addressing scale ambiguity using optical flow and priors. VPGS-SLAM [40] further scales 3DGS-SLAM to large scenes using a voxel-based progressive optimization strategy, balancing reconstruction fidelity and computational efficiency in outdoor and multi-room indoor environments.

SplaTAM [6] proposed a real-time pipeline that jointly optimizes camera pose and Gaussian parameters using only RGB inputs. Its global optimization strategy significantly mitigates cumulative drift. DG-SLAM [27] further incorporated dynamic label optimization for each Gaussian, distinguishing dynamic and static content based on temporal consistency. However, it primarily relies on single-frame analysis, lacking probabilistic temporal fusion.

Compared to MLP-based NeRF SLAM methods, 3DGS-based SLAM systems offer superior runtime efficiency, differentiable rasterization, and high visual fidelity at lower computational cost. The explicit Gaussian parameterization also allows easier integration with classical geometry-based methods and opens possibilities for semantic labeling, uncertainty modeling, and instance-level manipulation. Recent works such as DeRainGS [41] and LLGS [42] further showcase the potential of 3DGS in challenging environments like rain or extreme darkness, highlighting its versatility across visual degradation scenarios. However, 3DGS-based methods still face several challenges, particularly in dynamic environments where frame-to-frame inconsistency can lead to degraded optimization and fragmented map updates.

Most current 3DGS-SLAM approaches focus on optimizing static scenes. Although some frameworks introduce heuristic segmentation or per-frame classification to filter out dynamic content, they often fail to maintain temporal consistency or probabilistic confidence across views. This leads to unstable performance in real-world scenarios involving occlusions, articulated motion, or partial rigidity. Moreover, the lack of online dynamic reasoning severely limits their applicability in robotics, AR, and autonomous systems where live interaction with changing environments is required. Several recent methods, such as Dy3DGS-SLAM [43] and GARAD-SLAM [44], attempt to mitigate these limitations by introducing dynamic segmentation and anti-dynamic filtering mechanisms, while DenseSplat [45] demonstrates the benefits of neural priors in densifying sparse Gaussian maps, yet they still struggle with robust fusion of motion cues across time and views.

To address these limitations, BDGS-SLAM is proposed to improve the robustness and ensure the integrity of static scene reconstruction in 3DGS-SLAM under dynamic environments, effectively suppressing dynamic artifacts and preserving clean, high-quality maps. Our BDGS-SLAM addresses these limitations by introducing Bayesian filtering and multi-view probability updates to estimate dynamic probabilities, map them to Gaussians for label updating, and perform direct dynamic segmentation during rendering. This unified dynamic information flow improves tracking robustness and generates high-quality static scene maps in real time.

As 3DGS becomes more widely adopted, future SLAM frameworks may benefit from integrating probabilistic modeling, attention-based Gaussian filtering, and physically grounded motion priors. Such directions could further bridge the gap between explicit geometric SLAM and implicit neural modeling, enabling high-speed, high-fidelity, and dynamic-aware scene understanding.

## 3. Methodology

Our system, BDGS-SLAM, is designed to enable robust dynamic scene reconstruction and accurate camera tracking using a tightly coupled architecture that jointly optimizes semantic detection, probabilistic dynamic filtering, and rendering-aware Gaussian optimization. The overall architecture is shown in Figure 1. Unlike certain SLAM methods that rely on the static-world assumption and often treat tracking and mapping as relatively independent processes, BDGS-SLAM emphasizes a probabilistic framework with tighter integration between pose tracking and map optimization. In our system, dynamic/static label updates from the mapping module directly refine tracking, while tracking results immediately guide map updates, forming a closed-loop feedback mechanism.

### 3.1. System Overview and Motivation

The BDGS-SLAM pipeline consists of a tracking and mapping thread based on 3D Gaussian point representation. In this framework, the labeled Gaussian point cloud $P_{i}$ consists of points enriched with ORB features, rotation matrices, opacity α, anisotropic covariance Σ, and spherical harmonic coefficients for view-dependent appearance. These Gaussians are optimized jointly for both pose tracking and static scene rendering.

The system introduces three key probabilistic reasoning modules: (1) **Bayesian filtering-based dynamic verification**, (2) **multi-view probability fusion with temporal decay**, and (3) **adaptive dynamic Gaussian optimization**. Together, these modules ensure that dynamic interference is filtered with minimal loss of valid static structure, a critical balance in dynamic SLAM scenarios. We also embed a Kalman filter module for coarse motion prediction, enabling frame-to-frame initialization in highly dynamic scenes.

For clarity, we define the tracking module of BDGS-SLAM as the component responsible for extracting semantic and geometric features (YOLO detections and depth-based geometry), while the mapping module refers to the Gaussian mapping and Bayesian filtering that jointly estimate the trajectory and classify Gaussian points into dynamic or static categories. This distinction allows us to integrate learned priors from YOLO in the tracking stage with probabilistic consistency checks in the mapping stage, leading to robust performance in dynamic environments.

In designing this architecture, we deliberately prioritize a balance between accuracy, scalability, and runtime efficiency. While end-to-end neural implicit methods offer photorealistic outputs, their inference overhead remains prohibitive for real-time applications. By employing explicit Gaussian representations and tightly coupling semantic reasoning with optimization, BDGS-SLAM avoids redundant computations and maintains system responsiveness.

### 3.2. Bayesian Dynamic Inference on Gaussian Points

Traditional semantic SLAM schemes usually rely on tracking modules [12,13,14,46,47,48], to eliminate feature points associated with dynamic objects when dealing with dynamic objects, so as to suppress feature mismatching caused by dynamic objects. Then, dynamic objects are removed by pixel-level segmentation to ensure the accuracy of mapping. However, in SLAM systems based on 3D Gaussian distribution (3DGS), although pixel-level segmentation can effectively eliminate dynamic interference, its computational overhead is huge, and some static Gaussian points may be deleted by mistake, which will damage the quality of the constructed image. Conversely, without pixel-level segmentation, residual effects of dynamic objects may cause distortion in the final rendering result.

To solve this problem, we propose a probabilistic inference framework based on Bayesian filtering, The overall structure is shown in Figure 2. Different from the traditional way, we construct a Bayesian filter based on Gaussian points directly at the back end to classify dynamic and static Gaussian points. According to the time sequence, the algorithm backtracks and maps the classification results back to the tracking feature points, so as to correct the accumulated error of tracking-based pose estimation. This method effectively eliminates the interference effect in the rendering process.

In our system, YOLOv5 is employed in the tracking module to detect dynamic objects and provide semantic labels. Each Gaussian point is associated with a semantic probability derived from YOLO outputs (e.g., person, chair), which is then fused with its geometric attributes such as depth, 3D position, and covariance. Together, these form the feature vector $f_{t}$ for each point, which is used in the Bayesian update. Specifically, the semantic scores from YOLO are treated as observation likelihoods, while geometric features contribute to the motion and measurement models.

We perform probabilistic dynamic classification on Gaussian points via Bayesian filtering. Let $s_{t}\in \left\{\mathrm{static},\mathrm{dynamic}\right\}$ represent the latent state of a Gaussian point at time *t*. The posterior belief $\beta (s_{t})$ is recursively updated based on semantic detections and the state transition model:(1)$$\begin{array}{cc} \bar{\beta }\left(s_{t}\right) & =\int p\left(s_{t}|s_{t-1}\right),\beta \left(s_{t-1}\right),ds_{t-1}, \end{array}$$(2)$$\begin{array}{cc} \beta (s_{t}) & =\eta ,p\left(o_{t}|s_{t}\right),\bar{\beta }\left(s_{t}\right), \end{array}$$
where $p(o_{t}|s_{t})$ is the observation likelihood derived from YOLOv5 outputs, and η is a normalizing constant. The observation likelihood is approximated via sigmoid calibration:(3)$$p\left(o_{t}|s_{t}\right)=\sigma \left(w^{T}f_{t}+b\right),$$
where $f_{t}$ denotes the joint feature vector of each Gaussian point, and w,b are learned parameters. Specifically, $f_{t}$ is formed by concatenating (i) semantic features obtained from YOLOv5, such as object class confidence projected onto the Gaussian position, and (ii) geometric features inherent to the Gaussian representation, including position, covariance, and local appearance descriptors. This fusion enables the observation model to leverage both high-level semantic priors and low-level geometric cues when estimating the probability of a Gaussian being dynamic or static.

Additionally, to maintain temporal consistency, we apply exponential decay over time intervals:(4)$$\bar{\beta }\left(s_{t}=\mathrm{dynamic}\right)=e^{-\alpha \Delta t}\left(\beta \left(s_{t-1}\right)-0.5\right)+0.5,$$
where Δt is the temporal gap and α is the decay coefficient. This softens sudden transitions and favors stable state evolution.

Bayesian filtering provides a principled way to integrate prior beliefs and uncertain observations. Compared to deterministic masks, it offers a probabilistic margin for dynamic label estimation, particularly valuable when dealing with partial occlusion or low-confidence detections. However, filtering alone may still propagate noise, motivating our design of multi-view correction mechanisms.

### 3.3. Temporal Fusion via Multi-View Bayesian Updates

Although Bayesian filtering can effectively verify most dynamic Gaussian points, mislabeling may still occur. For example, dynamic Gaussian points moving along polar lines may be missed, or static Gaussian points located in marginal regions may be misinterpreted as dynamic points, resulting in label errors. In mapping and rendering, erroneous removal or retention of dynamic Gaussians can compromise the integrity of scene reconstruction or introduce visual artifacts.

To further enhance label robustness, we extend the Bayesian formulation across multiple keyframes, as shown in Figure 3. Given a current frame *t* and *N* co-visible keyframes $\left\{t_{k}\right\}$, we compute a fused posterior:(5)$$\beta^{\prime }\left(s_{t}\right)=\eta^{\prime }\prod_{k=1}^{N}\left[p{\left(o_{t_{k}}|s_{t}\right)}^{w_{k}}\right],\bar{\beta }\left(s_{t}\right),$$
where $w_{k}=\exp(-\alpha |t-t_{k}\left|\right)$ encodes temporal proximity and $\eta^{\prime }$ ensures normalization. To mitigate temporal inconsistencies, we apply majority voting across frames with a margin δ:(6)$$s_{t}=\left\{\begin{array}{cc} \mathrm{dynamic}, & \mathrm{if}\;\beta (s_{t})>0.5+\delta ,\\ \mathrm{static}, & \mathrm{if}\;\beta (s_{t})<0.5-\delta ,\\ \mathrm{uncertain}, & \mathrm{otherwise}. \end{array}\right.$$

This multi-view aggregation improves label confidence in complex environments with occlusions or intermittent motion.

Multi-view updates improve robustness but incur extra memory and computation. To address this, we cache co-visibility graphs and selectively subsample frames with high viewpoint diversity. In practice, viewpoint diversity is measured by the relative pose difference between candidate frames: frames are retained if their translation exceeds a distance threshold $\tau_{t}$ or if their rotation differs by more than $\tau_{r}$ degrees compared to existing keyframes. This ensures that the selected frames provide complementary perspectives rather than redundant observations.

### 3.4. Adaptive Optimization with Dynamic Penalization

As shown in Figure 4, Hard removal of dynamic Gaussians can disrupt the continuity of map structures, especially under segmentation errors. Instead, we propose a soft penalization strategy. The full loss is:(7)$$L_{\mathrm{total}}=\lambda_{\mathrm{psim}}L_{\mathrm{psim}}+\lambda_{\mathrm{dyn}}\sum_{g\in G_{D}}w_{g}\alpha_{g}^{2}+\lambda_{\mathrm{reg}}L_{\mathrm{reg}},$$
where $w_{g}$ denotes the importance weight of Gaussian *g*, and $\alpha_{g}$ is its opacity. The regularization term is defined as(8)$$L_{\mathrm{reg}}=\sum_{g}\left(∥\nabla_{\mu_{g}}I_{r}{∥}^{2}+{∥\Sigma_{g}∥}_{F}^{2}\right),$$
which encourages spatial smoothness by penalizing image-space gradient irregularities and large covariance magnitudes.

The photometric similarity loss is defined as:(9)$$L_{\mathrm{psim}}=\left(1-\gamma \right)\left|I_{r}-I_{gt}\right|+\gamma \left(1-\mathrm{SSIM}(I_{r},I_{gt})\right),$$
which combines pixel-wise error and structural similarity. (Note: a typographical error “1” in the previous draft has been removed for clarity.)

To stabilize the posterior estimation, we apply temporal smoothing of $\beta (s_{t})$ using an Exponential Moving Average (EMA):(10)$$\beta_{t}^{\mathrm{EMA}}=\rho \;\beta_{t}+\left(1-\rho \right)\;\beta_{t-1}^{\mathrm{EMA}},$$
where ρ is a smoothing factor controlling the balance between the current and past values.

Based on the smoothed posterior, we define a rendering mask:(11)$$M_{g}=I\left[\beta_{t}^{\mathrm{EMA}}<0.7\right],$$
which ensures that only reliable Gaussians contribute to supervision. In practice, $M_{g}$ gates the contribution of each Gaussian during rasterization and loss computation, so that unreliable points are softly suppressed instead of being immediately discarded.

This adaptive pruning avoids sharp drops in rendered quality and accommodates misclassified static regions. It strikes a balance between conservative map preservation and dynamic suppression. Moreover, when integrated with the Gaussian pyramid $\left\{GP_{i}\right\}$ (Section 3.5), the system refines global structure first and progressively improves details while controlling runtime cost.

### 3.5. Hierarchical Rendering and Pyramid-Based Training

Inspired by coarse-to-fine learning, we adopt a Gaussian pyramid $\left\{GP^{i}\right\}$ to optimize parameters from low to high resolution. For each level *i*:(12)$$\theta^{i}=\arg\min_{\theta }L\left(I_{r}^{i},GP^{i}\left(I_{gt}\right)\right),\;i=L,\ldots ,0.$$

We set adaptive opacity thresholds $\tau_{i}$ for early pruning:(13)$$\alpha_{g}^{i}\leftarrow \alpha_{g}^{i}\;\cdot \;I\left[\alpha_{g}^{i}>\tau_{i}\right].$$

This design stabilizes coarse geometry first and prevents overfitting to noisy fine-scale details. It also improves training efficiency and resource allocation.

BDGS-SLAM integrates multi-stage probabilistic filtering, dynamic-aware optimization, and hierarchical training into a tightly coupled 3DGS framework. By balancing robustness and tractability, our system achieves accurate tracking and realistic reconstruction in challenging dynamic environments.

## 4. Experiments

### 4.1. Experiment Settings

**Assessment indicators.** Attitude estimation is evaluated using the root mean square error (RMSE) and standard deviation (STD) of the absolute trajectory error (ATE) [49]. Before evaluation, Horn’s Procrustes method [50] was used to align the estimated trajectory with the true trajectory to ensure consistency of the evaluation benchmark. In our implementation, Horn’s method is restricted to compute only rigid rotation and translation, while keeping the absolute scale unchanged. Since BDGS-SLAM leverages RGB-D depth information, metric scale is inherently preserved, ensuring that the reported ATE values are valid and unaffected by scale correction. To evaluate the reconstruction quality of static maps in dynamic scenes, we obtain an approximate ground-truth (GT) of the static background using vision foundation models. Dynamic objects are first segmented, and the occluded regions are then filled by an inpainting model to generate a clean static background, which is used as proxy GT for computing peak signal-to-noise ratio (PSNR), structural similarity index (SSIM) and perceptual loss (LPIPS). These indexes were used to analyze the reconstruction results quantitatively from two dimensions: pixel similarity and semantic perception.

**Datasets.** The performance of the method was validated on three public and challenging datasets: the TUM RGB-D dataset [49], the BONN RGB-D dynamic dataset [51], and OpenLoris-Scene [52]. The selected datasets contain complex dynamic scenes and typical static environments, which help to comprehensively evaluate our method under diverse conditions and test its applicability and robustness in real indoor applications.

**Implementation details.** BDGS-SLAM runs on devices equipped with NVIDIA RTX 4090 GPU ( Manufacturer: NVIDIA Corporation, Santa Clara, CA, USA) (memory usage approx. 8GB) and processes approximately 2 frames per second (FPS) on the TUM, BONN, and OpenLoris-Scene datasets. The number of Gaussian pyramid levels is set to 3 (i.e., n=2), and the model is trained with loss weights of $\lambda_{psim}=0.85$ and $\lambda_{dyn}=0.15$. The iteration times of tracing and mapping are fixed at 20 and 40, respectively. This method is compared with Photo-SLAM [2], SplaTAM [6], DG-SLAM [27], ESLAM [29], and Co-SLAM [30].

### 4.2. Evaluation of the TUM RGB-D Dataset

The TUM RGB-D dataset [49] serves as a standard benchmark for evaluating SLAM systems in indoor environments, providing accurate ground-truth trajectories and diverse dynamic scenarios. It contains a variety of sequences involving both static backgrounds and dynamic elements, such as walking people, providing a rigorous setting for validating robustness against dynamic interference. In this experiment, we tested BDGS-SLAM against several state-of-the-art SLAM systems, including Photo-SLAM [2], SplaTAM [6], DG-SLAM [27], ESLAM [29], and Co-SLAM [30].

Table 1 presents the absolute trajectory error (ATE) and standard deviation (STD) results across various sequences. BDGS-SLAM consistently achieved the lowest ATE values, with an average of 0.0247 m across all sequences, significantly outperforming the baseline methods. For instance, in the sequence fr3/w_s_val, BDGS-SLAM achieved an ATE of only 0.0173 m, with a standard deviation of 0.0085 m, while other methods such as Co-SLAM and SplaTAM suffered from high trajectory drift. These results demonstrate that the proposed Bayesian filtering and multi-view probability update mechanisms effectively suppress dynamic noise and stabilize pose estimation in the presence of moving objects.

In terms of scene reconstruction quality, Table 2 reports the PSNR, SSIM, and LPIPS metrics. BDGS-SLAM outperforms all compared methods with a PSNR of 23.54 dB and SSIM of 0.900, indicating high-fidelity reconstruction. Notably, the LPIPS score (lower is better) is 0.159, which is substantially lower than that of DG-SLAM (0.186) and Photo-SLAM (0.264). Visual inspection of the rendered images shows that BDGS-SLAM retains fine structural details even in regions with high dynamic interference, avoiding ghosting artifacts commonly found in alternative methods.

Moreover, the improvement in both localization and rendering metrics confirms the synergy between the dynamic Gaussian verification module and the adaptive mapping strategy. Unlike traditional dynamic filtering approaches that operate only at the tracking stage, BDGS-SLAM adopts a tightly coupled mapping module, enabling continuous refinement of dynamic labels across frames. The exponential decay factor in the probability update ensures timely response to state changes, avoiding over-suppression or excessive preservation of Gaussian points. These mechanisms contribute to a robust and generalizable system suitable for real-world deployment in dynamic indoor environments.

### 4.3. Evaluation of the BONN RGB-D Dataset

The BONN RGB-D [51] dataset poses an elevated challenge for SLAM systems due to its highly dynamic content. It features sequences containing multiple fast-moving objects such as people, balls, and boxes interacting within indoor scenes. Unlike datasets with relatively static backgrounds, BONN emphasizes situations where large portions of the scene undergo rapid changes, making it ideal for testing the robustness of dynamic feature rejection and scene reconstruction fidelity.

BDGS-SLAM was evaluated on eight representative sequences from this dataset, and the results are summarized in Table 3 and Table 4. In terms of localization accuracy, BDGS-SLAM achieved the best performance across all sequences, with an average absolute trajectory error (ATE) of 0.0711 m and a standard deviation (STD) of only 0.0205 m. In particularly challenging sequences like syn2 and ball2, where traditional methods like SplaTAM and Photo-SLAM either failed or suffered severe drift, BDGS-SLAM maintained stable tracking with ATE values of 0.1439 m and 0.0303 m, respectively. These results underscore the effectiveness of our Bayesian motion inference and multi-view probability update in handling fast and complex object motion. For map reconstruction, BDGS-SLAM again demonstrated superior results. According to Table 4, our method achieves the highest PSNR (23.33 dB) and SSIM (0.940), while maintaining the lowest LPIPS score (0.295). This translates to clearer textures, better geometric consistency, and minimal artifacts. In contrast, even advanced dynamic-aware SLAM methods such as DG-SLAM exhibited ghosting and blurring in regions where dynamic objects moved rapidly across frames. Notably, BDGS-SLAM managed to reconstruct clean backgrounds behind occluded regions by leveraging temporal information and suppressing dynamic Gaussians.

As shown in Figure 5, qualitative visualization further illustrates the advantages of our method. For instance, in the p_tracking2 sequence, BDGS-SLAM maintains consistent reconstruction of the floor and background wall, while other methods either hallucinate false structures or leave holes due to incorrect dynamic handling. This robustness stems from the adaptive dynamic Gaussian optimization strategy, which penalizes but does not immediately discard uncertain regions, allowing for temporal re-evaluation.

In summary, the BONN dataset experiments affirm that BDGS-SLAM is capable of handling scenes with aggressive dynamics, ensuring both trajectory stability and map quality. These capabilities are critical for real-world robotic applications such as human–robot interaction or navigation in crowded environments, where dynamic obstacles are frequent and unpredictable.

### 4.4. Evaluation of OpenLoris-Scene RGB-D Dataset

The OpenLoris-Scene RGB-D dataset [52] presents an exceptionally challenging evaluation benchmark due to its focus on real-world, long-term, and dynamic human–robot interaction environments. Unlike the more controlled conditions of the TUM and BONN datasets, OpenLoris captures sequences in cluttered indoor settings such as offices, laboratories, and hallways, with frequent lighting changes, diverse dynamic objects, and non-repetitive motion trajectories. These properties make it ideal for assessing the generalization and stability of SLAM systems under real operational conditions.

BDGS-SLAM was tested on seven sequences from the office1 category, which includes a variety of dynamic interactions, such as people entering or leaving the scene, partially occluding the camera’s view, and environmental changes over time. As shown in Table 5, BDGS-SLAM achieved top-tier performance with an average ATE significantly lower than other methods. For example, in office1-3, BDGS-SLAM reached an ATE of only 0.043 m, while traditional systems such as Co-SLAM and ESLAM recorded errors more than twice as large. Moreover, in sequences like office1-4 and office1-7, where abrupt motion or persistent dynamic interference is present, baseline methods failed to maintain consistent tracking, resulting in fragmented or incomplete maps.

The superior performance of BDGS-SLAM on this dataset stems from its adaptive Gaussian optimization strategy and robust dynamic labeling. The exponential decay-based probability update mechanism adapts to changing motion patterns, while the delayed deletion policy for ambiguous Gaussian points ensures that critical scene components are not mistakenly discarded during short-term occlusions. These capabilities are crucial for OpenLoris, where dynamic elements often overlap with static structures, creating ambiguity that simpler filtering strategies cannot resolve.

Qualitative comparisons further validate the advantage of our method. For instance, in office1-6, BDGS-SLAM reconstructs the entire desk region with clean textures and coherent geometry, while Co-SLAM and SplaTAM suffer from ghosting and missing geometry in areas affected by human motion. Additionally, BDGS-SLAM preserves fine details like monitor edges and chair contours, which are often lost in other methods due to dynamic segmentation errors.

In conclusion, the experiments on the OpenLoris-Scene dataset confirm that BDGS-SLAM maintains strong generalization and robustness across diverse, unstructured real-world environments. Its capability to preserve map integrity in the presence of complex, overlapping dynamics indicates its practical suitability for long-term autonomous deployment in service robotics, surveillance, and smart environments.

### 4.5. Ablation Experiments

To rigorously evaluate the contribution of each core module in the BDGS-SLAM framework, we conducted a comprehensive ablation study on both the TUM RGB-D fr3/walking_xyz_val sequence and the BONN person_tracking sequence. Four configurations were tested: (1) without Bayesian filtering (w/o Bayefilter), (2) without multi-view probability update (w/o Multiview), (3) without Gaussian mapping (w/o Mapping), and (4) a full version with all modules (w/o Full indicating complete setup). In particular, the configuration “w/o Mapping” refers to disabling the Gaussian splatting mapping optimization. In this setting, the system still performs tracking with feature extraction and Bayesian dynamic verification but does not maintain or refine the global Gaussian map. Instead, only keyframe poses are optimized using geometric consistency, which allows the trajectory to be estimated but without constructing a consistent Gaussian scene representation. Therefore, this variant behaves similarly to a lightweight visual odometry baseline, and serves to highlight the contribution of the Gaussian mapping module to both trajectory accuracy and map quality. Each variant was evaluated in terms of absolute trajectory error (ATE) and standard deviation (STD), with the results summarized in Table 6.

The results demonstrate that each module significantly enhances the system’s performance. Removing Bayesian filtering leads to a marked increase in ATE, with the TUM dataset’s average error rising to 0.3562 m and STD to 0.1847. This shows that Bayesian inference is crucial for identifying dynamic Gaussians and correcting pose drift caused by transient motion. Without this module, the system is more prone to misclassifying dynamic objects as static, introducing artifacts in the map and reducing pose accuracy.

Eliminating the multi-view probability update mechanism (w/o Multiview) also causes a substantial decline in performance. In the BONN dataset, ATE increases to 0.3222 m compared to only 0.0182 m in the full version. This module enables temporal verification across co-visible keyframes, helping correct false labels and recover mistakenly removed static Gaussians. Its absence leads to persistent visual inconsistencies, especially in areas affected by fast motion.

The Gaussian mapping strategy (w/o Mapping) plays a pivotal role in refining the dynamic/static labeling. Disabling this module reduces the system’s ability to adaptively suppress residual dynamics during optimization. Although pose estimation remains relatively stable (TUM ATE = 0.0222 m), the rendered results become noisier, with more motion-induced artifacts and incomplete reconstructions, especially in cluttered dynamic scenes.

The full version of BDGS-SLAM, integrating all components, achieves the best balance of accuracy and robustness. The combination of dynamic Gaussian verification, multi-view update, and adaptive optimization enables the system to handle complex motion patterns, label ambiguities, and partial occlusions gracefully.

In summary, the ablation experiments verify the necessity of each component in BDGS-SLAM. Each module addresses specific failure modes—such as temporal inconsistencies, label misclassification, or residual dynamics—and only through their integration can the system maintain high-fidelity tracking and rendering performance in highly dynamic environments.

### 4.6. Time Consumption Analysis

To evaluate the computational efficiency of BDGS-SLAM, we conducted a runtime comparison against several baseline methods on the TUM RGB-D dataset, particularly focusing on the fr3/w_s sequence. All experiments were executed on an identical hardware platform equipped with an NVIDIA RTX 4090 GPU ( Manufacturer: NVIDIA Corporation, Santa Clara, CA, USA) and system configurations were standardized with 20 iterations for tracking and 40 iterations for mapping. The average runtime (in milliseconds per frame) for each system component was recorded and is summarized in Table 7.

BDGS-SLAM demonstrates competitive runtime performance compared to existing methods. Specifically, our tracking module achieves an average processing time of 81.4 ms per frame, which is the lowest among all evaluated methods, including DG-SLAM (89.2 ms), Photo-SLAM (223.6 ms), and SplaTAM (267.4 ms). This efficiency can be attributed to the use of lightweight Bayesian filtering for dynamic inference and a sparse optimization strategy in pose estimation, which avoids expensive dense computation or complex multi-scale processing.

In terms of mapping time, BDGS-SLAM exhibits a moderate computational cost at 473.4 ms per frame. Although this is slightly higher than SplaTAM (330.4 ms), it remains significantly more efficient than ESLAM (1641.4 ms) and DG-SLAM (549.3 ms), especially considering the additional overhead introduced by our multi-view probabilistic update and adaptive Gaussian optimization. The mapping thread includes temporal fusion, dynamic label refinement, and loss-driven Gaussian pruning, all of which are essential for achieving high-quality scene reconstruction in dynamic environments.

Importantly, our reported timings exclude the computational cost of the semantic segmentation module (YOLOv5), which runs asynchronously on a separate GPU thread. In practice, this modular separation allows BDGS-SLAM to maintain real-time capability with minimal interference from the semantic backend. Further acceleration is also possible through the use of lighter object detectors or pruning low-confidence dynamic labels in earlier stages.

Overall, BDGS-SLAM achieves a balanced trade-off between accuracy and efficiency. With an average total runtime of 565.3 ms per frame (corresponding to approximately 1.77 FPS), the system supports near real-time operation while delivering state-of-the-art performance in both pose tracking and dense photorealistic reconstruction. This makes it suitable for deployment in service robots, AR/VR devices, or other resource-sensitive real-time applications that operate in dynamic environments.

## 5. Conclusions

We propose BDGS-SLAM, a novel dynamic scene SLAM system based on 3D Gaussian Splatting, which effectively addresses the challenges of tracking drift and reconstruction artifacts caused by dynamic objects in real-world environments. The first contribution of our work is the integration of a Bayesian filtering framework into the tracking pipeline, enabling dynamic probability inference and reliable detection of transient moving points. Secondly, we introduce a multi-view probability update mechanism with exponential decay, which aggregates co-visible keyframe information to refine dynamic/static labeling and recover mistakenly removed static Gaussians. Thirdly, we design an adaptive optimization strategy that imposes penalizing constraints on dynamic Gaussians during mapping and rendering, thereby preserving the static scene structure while suppressing visual artifacts. Compared with baseline methods including Co-SLAM, ESLAM, SplaTAM, and Photo-SLAM, our BDGS-SLAM achieves substantial improvements in both trajectory accuracy and reconstruction quality across all datasets. Notably, even when compared to the most advanced baseline, DG-SLAM, our method consistently performs better: on the TUM dataset, BDGS-SLAM reduces the average ATE from 0.0441 m to 0.0247 m (a 43.9% improvement), and further boosts rendering quality by increasing PSNR from 22.34 dB to 23.54 dB, and SSIM from 0.799 to 0.900, and lowering LPIPS from 0.186 to 0.159. These results demonstrate that BDGS-SLAM not only surpasses earlier dynamic SLAM systems but also advances upon the current state of the art. In the future, we plan to enhance the system’s real-time capability and robustness by integrating lightweight semantic segmentation networks and developing memory-efficient modules for long-term dynamic scene understanding.

## Figures and Tables

**Figure 1 sensors-25-06641-f001:**
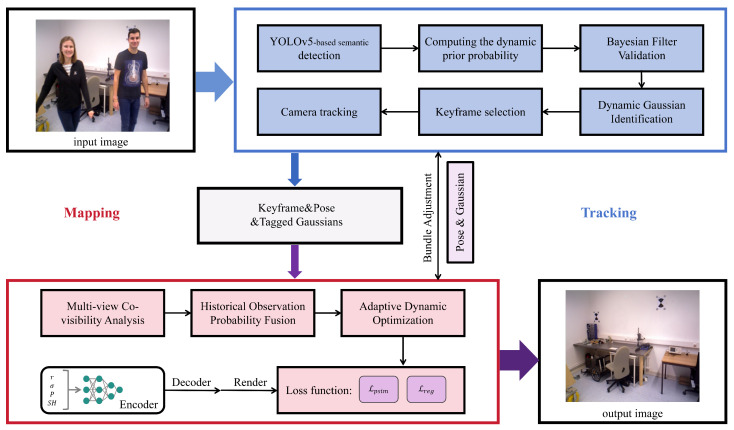
Architecture of BDGS-SLAM. The framework tightly couples tracking and mapping: YOLOv5 provides semantic priors in the tracking module; Bayesian filtering and multi-view updates in the mapping module refine dynamic/static labels; adaptive Gaussian optimization ensures robust reconstruction in dynamic environments.

**Figure 2 sensors-25-06641-f002:**
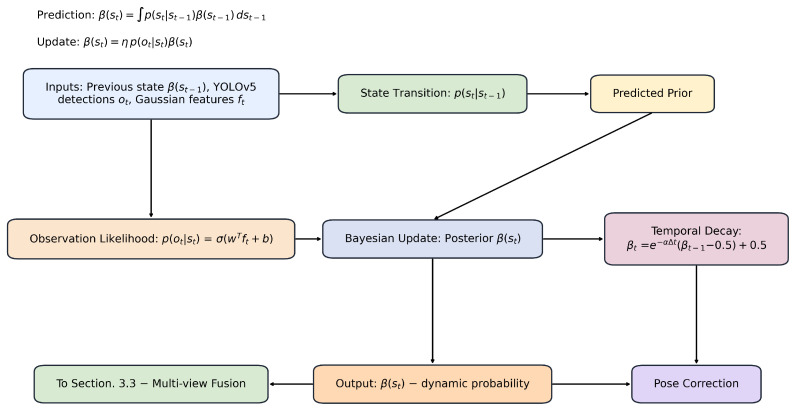
Bayesian State Update Framework. This figure illustrates the Bayesian state update in BDGS-SLAM. The inputs, including observations and prior state information, are processed through the state transition model. A Bayesian update is then applied to refine the posterior distribution of the state, ensuring robustness in dynamic environments.

**Figure 3 sensors-25-06641-f003:**
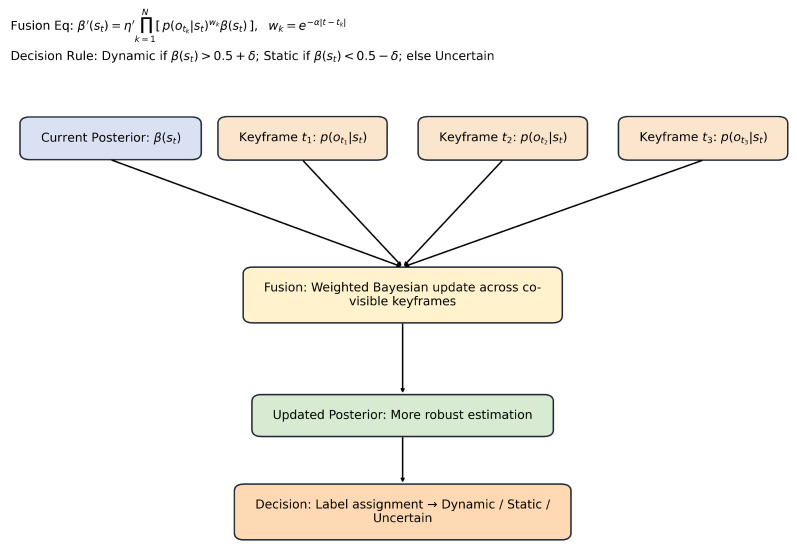
Multi-View Bayesian Fusion. This diagram shows how the current posterior and multiple keyframe observations are jointly fused. Each keyframe contributes through a weighted Bayesian update, where temporal weights reduce the influence of distant frames. The fusion produces an updated posterior that is more reliable, followed by a decision step that classifies points into dynamic, static, or uncertain categories.

**Figure 4 sensors-25-06641-f004:**
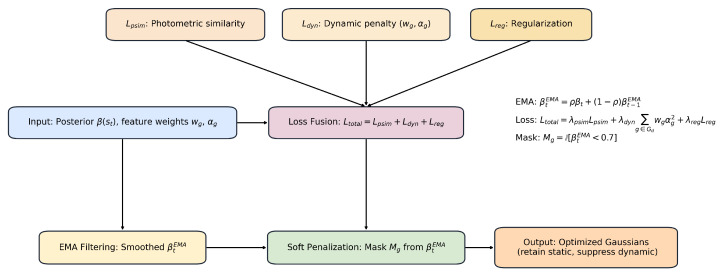
Adaptive Optimization with Dynamic Penalization. This figure depicts the adaptive optimization process. The posterior is smoothed using EMA filtering to reduce short-term fluctuations. Three loss terms—photometric similarity, dynamic penalty, and regularization—are fused into the total loss. A soft penalization mask derived from the smoothed posterior selectively downweights dynamic regions without discarding them. The final output is an optimized Gaussian representation that preserves static structures while suppressing dynamic components.

**Figure 5 sensors-25-06641-f005:**
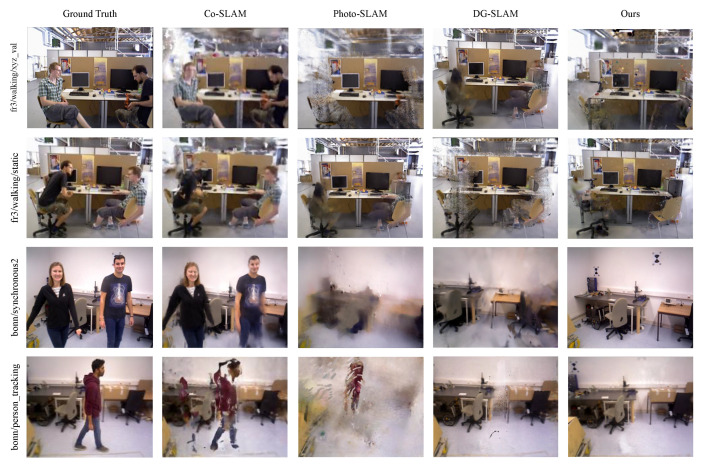
Qualitative comparison of rendering results on BONN RGB-D dynamic dataset. The figure illustrates that BDGS-SLAM generates fewer artifacts and maintains higher fidelity in static scene reconstruction compared to baseline methods (Photo-SLAM [2], DG-SLAM [27], Co-SLAM [30]), especially in handling complex dynamic objects.

**Table 1 sensors-25-06641-t001:** Results of ATE measurement on TUM dataset. The best performing results are highlighted as first, second, third. X “indicates tracing failure”. Units of measure are [m].

Sequences	Co-SLAM		ESLAM		SplaTAM		Photo-SLAM		DG-SLAM		Ours	
	ATE	STD	ATE	STD	ATE	STD	ATE	STD	ATE	STD	ATE	STD
fr3/w_x	0.6683	0.2684	0.6637	0.2652	0.4553	0.2919	0.2576	0.1671	0.0616	0.0233	0.0316	0.0147
fr3/w_x_val	0.7286	0.2111	0.6134	0.4748	0.7918	0.5054	0.5316	0.3725	0.0407	0.0182	0.0267	0.0095
fr3/w_s	0.5403	0.2771	0.5433	0.2523	0.4843	0.3171	0.1698	0.0230	0.0347	0.0247	0.0198	0.0146
fr3/w_s_val	0.4582	0.2365	0.2887	0.2056	0.8160	0.4114	0.1944	0.0835	0.0239	0.0191	0.0173	0.0085
fr3/w_r	2.2456	1.0751	0.9870	0.4961	0.0831	0.0376	0.5204	0.2956	0.0523	0.0263	0.0269	0.0137
fr3/w_r_val	2.2658	0.8892	0.9580	0.3017	1.2746	0.7962	0.2606	0.1868	0.0541	0.0196	0.0271	0.0129
fr3/w_h	0.8056	0.3922	0.7691	0.5303	0.8053	0.5508	0.2256	0.1081	0.0383	0.0149	0.0236	0.0074
fr3/w_h_val	0.6556	0.4013	0.9371	0.1838	1.2568	0.9423	0.3834	0.1843	0.0470	0.0228	0.0245	0.0172
Avg.	1.0460	0.4689	0.7200	0.3387	0.7459	0.4816	0.3179	0.1776	0.0441	0.0211	0.0247	0.0123

**Table 2 sensors-25-06641-t002:** Results of PSNR↑, SSIM↑, and LPIPS↓ on the TUM RGB-D fr3/walking sequences (w denotes walking). Columns correspond to the sub-sequences fr3/w_xyz, fr3/w_xyz_val, fr3/w_s, fr3/w_s_val, fr3/w_rpy, fr3/w_rpy_val, fr3/w_half, and fr3/w_half_val. Best results are highlighted as first, second, and third.

Methods	Metrics	Avg.	xyz	xyz_val	s	s_val	rpy	rpy_val	half	half_val
cCo-SLAM	PSNR↑ [dB]	8.22	8.68	8.73	11.15	3.70	6.54	8.03	9.04	9.86
	SSIM↑	0.503	0.386	0.557	0.435	0.752	0.374	0.466	0.491	0.563
	LPIPS↓	0.588	0.733	0.514	0.709	0.356	0.472	0.526	0.768	0.629
ESLAM	PSNR↑ [dB]	8.02	10.10	6.90	10.08	7.04	4.75	4.12	9.79	11.37
	SSIM↑	0.502	0.402	0.625	0.446	0.506	0.546	0.473	0.612	0.403
	LPIPS↓	0.537	0.677	0.462	0.680	0.430	0.502	0.463	0.566	0.515
SplaTAM	PSNR↑ [dB]	10.29	11.83	12.30	12.32	9.94	6.32	8.46	9.27	11.84
	SSIM↑	0.574	0.641	0.551	0.638	0.492	0.530	0.579	0.592	0.568
	LPIPS↓	0.371	0.424	0.291	0.374	0.365	0.454	0.348	0.381	0.330
Photo-SLAM	PSNR↑ [dB]	13.79	16.30	15.70	11.31	12.92	12.17	12.51	14.93	14.45
	SSIM↑	0.597	0.442	0.652	0.657	0.584	0.616	0.558	0.648	0.618
	LPIPS↓	0.264	0.277	0.228	0.340	0.223	0.322	0.274	0.195	0.254
DG-SLAM	PSNR↑ [dB]	22.34	23.63	20.90	24.70	20.78	23.07	21.63	18.33	25.71
	SSIM↑	0.799	0.826	0.981	0.710	0.775	0.860	0.633	0.847	0.763
	LPIPS↓	0.186	0.221	0.136	0.247	0.133	0.185	0.154	0.204	0.205
Ours	PSNR↑ [dB]	23.54	20.64	24.86	25.17	21.00	24.97	21.76	24.88	25.01
	SSIM↑	0.900	0.885	0.879	0.890	0.912	0.809	0.978	0.919	0.928
	LPIPS↓	0.159	0.237	0.269	0.153	0.094	0.112	0.093	0.184	0.130

**Table 3 sensors-25-06641-t003:** Evaluation of BONN RGB-D dataset. The best performing results are highlighted as first, second, third. X “indicates tracing failure”. Units of measure are [m].

Sequences	Co-SLAM		ESLAM		SplaTAM		Photo-SLAM		DG-SLAM		Ours	
	**ATE**	**STD**	**ATE**	**STD**	**ATE**	**STD**	**ATE**	**STD**	**ATE**	**STD**	**ATE**	**STD**
crowd	0.5127	0.3742	0.5164	0.0995	2.0423	1.0411	0.4800	0.3338	0.0823	0.0441	0.0440	0.0145
crowd2	0.6372	0.3292	0.5382	0.2354	3.4029	1.6681	0.9896	0.5448	0.0917	0.0511	0.0483	0.0274
p_tracking	0.8428	0.5203	0.6570	0.2986	0.2603	0.2913	0.5429	0.2708	0.0473	0.0223	0.0403	0.0202
p_tracking2	0.7341	0.4498	0.7154	0.3188	0.5271	0.4891	0.6192	0.2968	0.0202	0.0105	0.0383	0.0208
syn	0.9148	0.4074	0.7684	0.5606	X	X	0.7951	0.4836	0.2088	0.0436	0.1983	0.0413
syn2	0.9036	0.4577	0.8679	0.6112	X	X	1.4931	0.4397	0.1936	0.0932	0.1439	0.0097
ball	0.6346	0.3007	0.5789	0.2957	2.3099	0.7633	0.0863	0.0366	0.0356	0.0111	0.0256	0.0095
ball2	0.7914	0.3187	0.4620	0.2433	1.3592	0.0687	0.2242	0.0686	0.0397	0.0323	0.0303	0.0205
Avg.	0.7464	0.3948	0.6975	0.3329	1.6503	0.7203	0.6538	0.3093	0.0899	0.0385	0.0711	0.0205

**Table 4 sensors-25-06641-t004:** Results of PSNR↑, SSIM↑, and LPIPS↓ on the BONN dynamic RGB–D dataset. Columns correspond to the sequences crowd, crowd2, p_tracking, p_tracking2, syn, syn2, ball, and ball2. Best results are highlighted as first, second, and third.

Methods	Metrics	Avg.	crowd	crowd2	p_tracking	p_tracking2	syn	syn2	ball	ball2
Co-SLAM	PSNR↑ [dB]	8.24	9.91	8.65	9.10	8.60	6.59	5.84	9.06	8.17
	SSIM↑	0.468	0.549	0.423	0.517	0.461	0.450	0.381	0.488	0.471
	LPIPS↓	0.610	0.566	0.571	0.624	0.599	0.697	0.657	0.643	0.522
ESLAM	PSNR↑ [dB]	8.38	10.11	8.85	8.73	8.98	6.49	6.34	8.20	9.37
	SSIM↑	0.506	0.561	0.514	0.513	0.460	0.495	0.499	0.506	0.496
	LPIPS↓	0.587	0.565	0.478	0.612	0.493	0.637	0.784	0.633	0.495
SplaTAM	PSNR↑ [dB]	9.16	10.32	9.77	9.79	8.86	7.38	7.41	10.44	9.32
	SSIM↑	0.521	0.577	0.515	0.554	0.524	0.504	0.440	0.589	0.465
	LPIPS↓	0.516	0.523	0.499	0.549	0.413	0.604	0.520	0.521	0.499
Photo-SLAM	PSNR↑ [dB]	10.52	12.64	10.63	10.93	10.77	8.51	9.06	10.56	11.05
	SSIM↑	0.671	0.718	0.702	0.691	0.684	0.669	0.588	0.686	0.631
	LPIPS↓	0.425	0.401	0.321	0.474	0.361	0.475	0.478	0.493	0.393
DG-SLAM	PSNR↑ [dB]	22.11	21.01	21.07	22.33	21.70	21.93	24.09	21.29	23.48
	SSIM↑	0.866	0.874	0.836	0.839	0.779	0.857	0.934	0.963	0.844
	LPIPS↓	0.325	0.389	0.294	0.363	0.322	0.349	0.256	0.352	0.272
Ours	PSNR↑ [dB]	23.33	21.78	22.41	22.98	22.91	23.55	24.92	23.65	24.44
	SSIM↑	0.940	0.864	0.972	0.927	0.936	0.951	0.969	0.947	0.950
	LPIPS↓	0.295	0.338	0.274	0.310	0.319	0.293	0.281	0.288	0.258

**Table 5 sensors-25-06641-t005:** OpenLoris-Scene. The best performing results are highlighted as first, second, and third.

Sequences	Co-SLAM		ESLAM		SplaTAM		Photo-SLAM		DG-SLAM		Ours	
	**ATE**	**PSNR**	**ATE**	**PSNR**	**ATE**	**PSNR**	**ATE**	**PSNR**	**ATE**	**PSNR**	**ATE**	**PSNR**
office1-1	0.109	15.12	0.125	17.89	0.093	22.28	0.084	23.02	0.072	23.17	0.061	24.91
office1-2	0.147	14.73	0.169	18.32	0.129	21.56	0.103	23.54	0.081	23.83	0.067	25.37
office1-3	0.040	15.54	0.073	17.53	0.064	22.67	0.055	23.14	0.042	23.48	0.027	25.72
office1-4	0.234	14.56	0.138	18.61	0.122	21.82	0.102	22.76	0.113	23.04	0.092	24.58
office1-5	0.131	15.33	0.163	17.94	0.131	22.51	0.123	23.32	0.112	23.81	0.103	25.14
office1-6	0.101	14.92	0.132	18.16	0.113	21.93	0.102	23.27	0.091	22.95	0.082	25.53
office1-7	0.243	15.07	0.181	18.05	0.156	22.21	0.121	23.39	0.101	23.51	0.090	25.26
Avg.	0.144	15.04	0.140	18.07	0.115	22.14	0.099	23.21	0.087	23.40	0.075	25.22

**Table 6 sensors-25-06641-t006:** Ablation study results for TUM and BONN datasets. Metric units IS [m]. The best performing results are underlined for emphasis.

Dataset	Metric	w/o Bayefilter	w/o Multiview	w/o Mapping	w/o Full
TUM	ATE	0.3562	0.2645	0.0222	0.0192
	STD	0.1847	0.1161	0.0132	0.0120
BONN	ATE	0.6570	0.3222	0.0285	0.0182
	STD	0.3336	0.1764	0.0153	0.0127

**Table 7 sensors-25-06641-t007:** Runtime comparison FPS [ms]↓ on TUM fr3/w_s. The best performing results are underlined for emphasis.

Methods	Co-SLAM	ESLAM	SplaTAM	Photo-SLAM	DG-SLAM	Ours
Tracking↓	101.4	2045.9	267.4	223.6	89.2	81.4
Mapping↓	470.1	1641.4	330.4	360.7	549.3	473.4
Avg. Running↓	571.6	3688.5	598.4	585.3	645.9	565.3

## Data Availability

No new data were created in this study. All datasets analyzed are publicly available from their original sources [TUM RGB-D, BONN RGB-D, OpenLoris-Scene RGB-D].

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
