# Peer review of "BDGS-SLAM: A Probabilistic 3D Gaussian Splatting Framework for Robust SLAM in Dynamic Environments"

_sensors, 2025, doi:10.3390/s25216641_

Round 1
Reviewer 1 Report
Comments and Suggestions for Authors
It is a good idea to filter out the dynamic objects and keep good reconstruction for surroundings with gaussian splating. I suggest the author make the code open-source,and have the following suggestiongs:
- In section 3.2, the formulation of Bayesian is still not clear, how to implement it in this algorithm may need a figure to illustrate this process. Same problems in Section 3.3 and 3.4
- The reconstruction results of the scene should be shown in 4.4
- The Ablation experiment is for one dataset? or for all datasets? This should be clear.
Reviewer 2 Report
Comments and Suggestions for Authors
The manuscript is generally clear, but certain sections would benefit from simplification and clearer explanations to improve readability. Figures and tables need higher resolution and more descriptive captions to better convey the findings. The methodology is not sufficiently detailed for replication; additional parameters and stronger justification of the chosen methods are necessary. The results and discussion section is too brief and should be expanded with comparisons to recent studies in order to emphasize the novelty and contribution of the work. Some references are outdated and require replacement with more recent sources, while the formatting of references also needs to be standardized. Finally, the paper contains minor language errors, so careful proofreading is recommended before publication.
Reviewer 3 Report
Comments and Suggestions for Authors# BDGS-SLAM
## Summary
The authors propose a method for GS-based SLAM in environment with dynamic objects. The method is based on using a Bayesian filtering.
Summary of the comments to the authors:
The paper is an interesting read and an interesting method/research direction. I have enjoyed learning about the subject and I think it will make a good contribution in the future. However, there are questions that must be addressed before that happens. See my detailed comments under this but the main questions are:
1. Why is Horn's method used to transform the trajectory before evaluation? AFAIK it also modifies the scale which would invalidate the results
2. The method description must be clarified. How is Yolo used exactly? Which feature are used? How are they integrated in the method? What is the back/front end?
3. Some of the notation in the equations can be clarified.
4. How can the GS backend be removed in the ablation tests?
5. The related work on "traditional" SLAM can be clarified and made more specific. In particular the opposition between traditional and dynamic SLAM is (IMO) incorrect.
## Questions
### Introduction
* page 2: What is the "rendering artifact problem caused by dynamic noise input". Unclear why traditional methods cannot be applied since they are based on semantics. Actually, DN-SLAM and DDN-SLAM use traditional strategies.
* page 2: line 50-55 are written in a way that is not easy to follow. the "Mapping module: XX" format is harder to read than a normal sentence.
### Related Work
* "traditional SLAM systems suffer from a fundamental limitation: the static scene assumption. They treat all observations as originating from a rigid, unchanging world." This is untrue. Traditional SLAM systems able to handle dynamic object exist (as noted by the authors in the introduction, often using semantic filtering). For example <https://github.com/Horacehxw/Dynamic_ORB_SLAM2?tab=readme-ov-file>. After having read the second section "Dynamic SLAM", I think the definition of traditional SLAM and it's limitation is circular -> Traditional SLAM are SLAM methods assuming a static environment and so a limitation is that it cannot handle dynamic objects.
* "these methods rely purely on geometric consistency and ignore high-level semantics or learned priors." While this is a common problem I think the statement is too strong. Method using prior information exist <https://www.mdpi.com/2218-6581/8/2/40>
* "However,it" missing a space
* " for dynamic scenes.DG-SLAM" Missing a space.
* Section 2.3, I'm not sure why the whole multi-agent paragraph is relevant to this paper.
* line 236: what does "map purity of 3DGS-SLAM" mean?
### Methodology
* Figure 1 font is too small and very hard to read with the colors. In general, this image must be improved.
* "In contrast to traditional decoupled SLAM pipelines, our system integrates map updates and pose tracking" I may not understand what the authors mean but SLAM pipeline usually couple map updates and pose tracking since it stands for Localization AND Mapping
* "we construct Bayesian filter based on Gaussian points directly at the back end to classify dynamic and static Gaussian points" at that point, the back VS front end is not explained as a concept so I cannot know what this means. I think the idea of back VS front end with respect to the authors method (i.e. GS is in the "backend" and Yolov5 is in the frontend?) need to be presented in a clearer manner
* "we perform" missing uppercase
* I would appreciate a more specific explanation of that the "the observation likelihood from YOLOv5 outputs" is at line 292. Especially, " ft denotes the semantic and geometric feature vector of the Gaussian point" points at the semantic feature being obtained from Yolo but it is unclear how they are fused with the geometric features in on element.
* "selectively subsample frames with high viewpoint diversity" How is this done?
* Eq 7, what is $w_g$?
* In Eq8, is the "1" an indicator function or is there missing parenthesis around the second term?
* What does "EMA" stand for?
* Unclear what the rendering mask is for because it is never explained how it is used in the method.
* "a Gaussian pyramid {GPi}" do you maintain multiple version of the map or do you slowly increase the resolution? Since you care about computational cost (cf related work) what is the impact of this on the computational cost?
Having finished reading the methodology section, I do not know how YOLO is used in the paper. I understand that there are semantic labels but when are they queried, which ones are we looking for, and how are they used is unclear to me. I think this part can be clarified and more details can be provided.
### Experiments
* I am _very_ confused by the authors' use of the Horn Procrustes method to align the estimated trajectory with the true trajectory. AFAIK, that method find the translation, rotation, and _scale_ between two set of points given their coordinates. While finding the translation and rotation is acceptable, the scale shouldn't be changed and would invalidate the results presented. Instead the authors should align the first pose of each trajectory and measure the difference using the ATE and the RPE
* "the true value of static background (GT) with the help of large models", how is this done?
* l364 "= 0.85and" missing a space
* Table 2 use midrule between all method to separate the sections
* l422 "Qualitative visualization further illustrates the advantages of our method." please add images to the paper to showcase it.
* Table6, why is openloris not used?
* In the ablation study it is unclear to me how you can test "without back-end Gaussian mapping (w/o Mapping)" How can the GS module be removed? Is it classic visual SLAM then?
* "time consumption analysis" Missing uppercase
Round 2
Reviewer 1 Report
Comments and Suggestions for Authors
I suggest the newly added figures should be clear and the size of the fonts should be larger.
Reviewer 3 Report
Comments and Suggestions for Authors
Thank you for the updated version, this is much clearer but I would like to suggest some more improvements and possible questions:
* This idea of front/back-end is still very confusing and not usual SLAM terminology. In this paper, the authors use it to mainly designate tracking (front end) VS mapping (back-end). Why not simply use tracking and mapping as in the vast majority of other SLAM papers? It makes the paper harder to understand, without bringing anything of value IMO.
* line 263 > "Unlike many traditional SLAM pipelines, where tracking and mapping are only loosely coupled and map updates provide limited feedback to the front-end, our BDGS-SLAM framework tightly integrates pose tracking and map optimization." -> I have read the addition to the related work and I am not at all convinced (from the related work and from my experience) that this statement about the traditional SLAM pipeline is true. In the related work the authors state that "therefore, rather than a strict opposition between “traditional” and “dynamic” SLAM, we adopt the terminology here to emphasize whether a method explicitly models dynamic objects or relies on the static-world assumption. " then why bring up traditional SLAM again instead of static-world assumption SLAM models? I actually think that a lot of SLAM method work as closed-loop system as described by the atuhors instead of a static-world model (e.g. NDT maps, GMapping, Dynamic SLAM...). The method in this paper as a probabilistic framework is interesting but the scope stated here is over-selling it.
* Fig 1 is more readable but I would wish the font in the flowchart itself to be bigger for people with disabilities.
* "For clarity, we define the front-end of BDGS-SLAM as the module responsible for extracting semantic and geometric features (YOLO detections and depth-based geometry), while the back-end refers to the Gaussian mapping and Bayesian filtering that jointly estimate the trajectory and classify Gaussian points into dynamic or static categories. This distinction allows us to integrate learned priors from YOLO in the front-end with probabilistic consistency checks in the back-end, leading to robust performance in dynamic environments" -> Thank you for this add-on! It is much clearer. However, as per my first comment, front and back end are mentioned earlier in the paper and it is thus hard to understand before this.
* Fig2, Fig3, Fig4 font size _in_ the flowchart is too small, it is not possible to read it.
* "In our system, YOLOv5 is employed in the front-end to detect dynamic objects and provide semantic labels. Each Gaussian point is associated with a semantic probability derived from YOLO outputs (e.g., person, chair), which is then fused with its geometric attributes such as depth, 3D position, and covariance. Together, these form the feature vector ft for each point, which is used in the Bayesian update. Specifically, the semantic scores from YOLO are treated as observation likelihoods, while geometric features contribute to the motion and measurement model" -> thank you this is much clearer!
